# The Application of a Sonic Probe Extensometer for the Detection of Rock Salt Flow Field in Underground Convergence Monitoring

**DOI:** 10.3390/s21165562

**Published:** 2021-08-18

**Authors:** Zbigniew Szczerbowski, Zbigniew Niedbalski

**Affiliations:** 1Faculty of Mining Surveying and Environmental Engineering, AGH University of Science and Technology, al. A. Mickiewicza 30, 30-059 Kraków, Poland; 2Faculty of Civil Engineering and Resource Management, AGH University of Science and Technology, al. A. Mickiewicza 30, 30-059 Kraków, Poland; zbigniew.niedbalski@agh.edu.pl

**Keywords:** monitoring of underground excavations, extensometer, salt flow, tectonic stress

## Abstract

Special regulations have been laid down to establish the principles and requirements for the safety and serviceability of old mining workings which are adapted for tourism. To comply with these regulations the measurements were taken in the Bochnia Salt Mine, which has been in use for 800 years. The presented work demonstrates the use of a sonic probe extensometer in connection with the obtained results of displacement measurements in intact rocks surrounding the gallery. There were also test measurements carried out for determination of the real accuracy of the instrument. The presented study of deformations detected by electromagnetic extensometer measurements is presumed to be the first time that research has been made in salt mines operating in rock mass affected by tectonic stress. The paper presents the process of rock salt flow into the gallery observed over a period of 3 years. It is an unprecedented depiction of salt deformation subjected to natural stresses. One of the more surprising results presented here is the discovery of the occurrence of a specific distribution of strain around the measured gallery. The results of measurements showed that the southern part of the intact rock mass surrounding the passage is more compressed (strain rate 3.6 mm/m/year) than the northern one (strain rate 1.6 mm/m/year). This illustrates the presence and influence of additional tectonic effects resulting from the Carpathian push. These observations represent a new kind of research into tectonic stress and tectonic activity in underground measurements.

## 1. Introduction

Conducting the appropriate monitoring of technical facilities is one of the most important activities affecting the safe use of these facilities and also their long-term maintenance. Depending upon the complexity of the facility and its intended use, the monitoring may be only visual, or it may be carried out with the use of survey equipment with the periodic or continuous recording of a parameter [1,2,3]. This problem of monitoring is of particular importance when the object under observation is an underground mine. In a situation like this there is a need to monitor underground workings to ensure the safety of personnel working in the mine and also to ensure the safety of the surface infrastructure. There are methods used to assess the phenomena occurring inside the rock mass [4,5] and to note any expected changes which may occur on the terrain surface as well as to study any terrestrial deformations [6,7,8]. Geodetic methods applied in monitoring of underground excavations are as follows [9,10,11]:-leveling for the estimation of vertical displacements. There are usually particular terrain and underground networks. Highly accurate observations of benchmarks reveal displacements important for the estimation of potential hazards;-horizontal observations. Distance and angle measurements are carried out on observational lines comprised of control points. Evaluated changes in their coordinates are used in the estimation of orientations and rates of displacement. The accuracy of the measurements is lower than that of leveling;-observations of mine shaft deformations. Measurements are devoted to deformation of a shaft’s axis and the verticality of shaft guides, or the measurements are based on control points mounted in the shafts, so they reveal vertical and horizontal displacements of the points. The points form an extraordinary spatial network that can be used for the monitoring of a large portion of rock mass deformations;-convergence surveying of underground excavations. The surveys are carried out for the design of the protection of underground excavations. There is a specialized engineering service involved in measurements: either stationary, which uses tubular, rod, tape or wire convergometers (for deformation measurements of bases in caverns, headings, pillars), or portable, which uses a spring tool for reading the measurements made using typical measuring tape (wire) or the even more popular laser rangefinders.

Monitoring in underground excavations is used for current evaluations of stability of roof strata or elements of support. Observations are conducted over the short term—a few months or a few years. These are the most common existing excavation periods. Nowadays for current monitoring in underground mines, instrumented bolts, sonic extensometer, tell-tale extensometer, fiber optic sensors, fiber Bragg grating, and stress meters are used [12,13,14,15,16].

The principle of measurement and the use of a magnetic extensometer has a long history but there are not many studies in the scientific literature devoted to the applications of these devices. This results probably from the fact that the instrument is usually applied for common practical engineering purposes. The most popular applications are geotechnical monitoring of dams [17,18], in tunneling to monitor the effect of tunnel construction on surrounding infrastructure [19], and in mining to monitor the stability of underground workings during mining operations, usually used for a short period of time [12,20].

Basically, so far no underground measurements have been carried out for long-term monitoring of displacements with the use of the instrument which have been discussed in the professional literature.

When the activities of a mine have ceased, monitoring is carried out for a certain period of time. This takes place mainly in the fields of hydrogeological and topographical changes and occurs continuously until the mine is completely closed down. However, there are extraordinary situations to be encountered when the mining plants have been operating continuously for several centuries [10,21]. This is the case with the Bochnia Salt Mine, which has been operating for nearly 800 years as a mine and for several decades as a museum. It has been listed as a UNESCO World Heritage Site and also as a Historic Monument of Poland. The necessity to monitor the geo-mechanical phenomena in the underground workings is important because the development of the town of Bochnia took place directly above the mine over a period of more than seven centuries [22]. Any loss of excavation stability within the mine could lead to a major construction disaster at ground level [23,24]. Previous observations have shown that the analyzed region is also affected by geodynamic activity.

Current research on deformations of the old excavations of the Bochnia Salt Mine aims to assess the safety of the underground tourist route, which runs through the historical passages and chambers of this mine, which is the property of the oldest and still active industrial companies in the World. All the available data deformation relating to measurements are rooted in the traditions of old surveying activities, the aim of which was to map the workings of a dynamically developing mine in the 16th century [10].

It is important to determine the stability of the workings by observations of any manifestation of rock mass pressure—a centuries-long impact resulting in the deformation of the original contours of the galleries and the chambers. Leveling and surveying tape measurements allow the obtaining of data on the convergence, manifested as the result of a reduction of excavation contour. Therefore, the most important element of this study is the analysis of the results of the deformation measurements which have been taken to determine the state of stress and strain in the rock mass in the vicinity of the excavations. Although the classical mine surveying measurements provided us with extensive, high quality observational material of the deformation parameters, we do not know enough about the propagation of deformation in salt rock mass. At present the stress and strain models that we have been given are only theoretical [10,11].

Jeremic presented three characteristics of salt deformations: the elastic, the elastic–plastic and the plastic behaviors of rock salt [25]. The elastic behavior of rock salt is assumed to be linearly elastic with brittle failure. The rock salt is observed as linear elastic only for a low magnitude of loading. In relation to this feature of rock salt, it is an essential responsibility of the authority of every salt mine to control the deformation and strains particularly, because at certain strain levels a plastic deformation takes place and the salt structure loses its strength, leading to sudden failure.

This paper aims to present the results of the taking of electronic measurements of the internal displacements of rocks surrounding the mine’s working. A dozen measurement series by sonic probe extensometer carried out over a period of three years provided data of the movements of points up to a distance of approx. 7 m from the contours of the passages at the entrance to the Saint Kinga Chapel. The displacement rates are very slow, but an appropriate time interval has been agreed upon which allows for the drawing of consistent conclusions.

## 2. Environmental Background of the Site

The most distinguishing feature of the site is its geological position—at the contact zone of two large tectonic units: the Carpathian Foredeep Basin and the Outer Carpathians (Figure 1). The shape of the salt deposit was formed in response to the forces and mechanisms that control salt flow and it reflects its turbulent geological history [26]. During the Late Miocene epoch, evaporite formations were folded in front of the Carpathian nappes, and then they were moved northward and thrust from the south over the autochthon [27,28,29,30]. Evaporate formations occur there as a belt of about 12 km in length along the border of the Carpathian overthrust, and the maximum width of the deposit is 200 m [27,31].

In the Bochnia region there is a boundary between the allochthonous (disturbed) and the autochthonous (non-disturbed) parts and the salt deposit morphology results from the work of tectonic force. The Bochnia salt deposit in the strict sense is formed as the Bochnia anticline in a lentoid shape (Figure 2). So the modern image of a complicated geological formation allows one to be aware of the role of tectonic forces acting in the area that formed the salt deposit.

The depicted rock mass deformations observed underground by geological methods [31] and the characteristic of the deformation process determined by geodetic measurements [9] can be indicative of recent neo-tectonic activity. The application of a sonic probe extensometer to measure the orogenic movements can be helpful in evaluating these phenomena.

Measurements of displacements resulting from rock mass movements induced by old mine operations (compression of old excavations) have been carried out for about one hundred years. In the beginning there were only leveling measurements of underground network points. Nowadays, leveling and convergence measurements are the basic ones, so benchmarks, control points and baselines are measured usually once a year.

Horizontal and vertical convergence are measured as changes of baseline distance are determined by flat steel tapes, steel bands and most recently by electronic distance measuring equipment. The area of the measurement is in the central part (in both horizontal and vertical projection) of the deposit between two shafts: The Campi and the Sutoris, which are connected by a longitudinal gallery—the August. This area is situated on the first mine’s level which was founded at the beginning of the 18th century. It is situated at a depth of 212 m below the surface, reaching a length of 3 km and it passes through the most impressive chamber—The Saint Kinga Chapel (Figure 3). The measurement site is located at the entrance to the chapel and has existed for over 20 years as a site providing for convergence measurements. The workings in this part of the mine are protected by wooden props only in certain sections and the side walls in the August passage at the chapel, in fact, are not supported.

The St. Kinga chapel is located in a chamber which dates back to the 18th century. It is the largest in size of the chapels which exist in the mine and its total area size is about 260 m^2^. Its current design dates back to the beginning of the 20th century (Figure 3).

The results of measurements which have been carried out there demonstrate a stable process of deformation which is differentiated in particular areas of the chapel. Regular measurements of chamber deformation have been carried out since 1993. In addition, there are significant and visible cracks which result from the pressure of salt rock mass. All the measurements are carried out by personnel of The Mine Surveying Department of the Bochnia Salt Mine.

The vertical displacements of the roof benchmarks in this area are −15 mm per year and the floor benchmarks are about −10 mm. This makes for a vertical convergence of an average value of about −5 mm and the average annual vertical strain rate is about 1.5%. These correspond to other results of measurements in this part of the mine (all benchmarks in this part of the mine have been showing the same displacements for years).

Within the section of the mine which was being measured by our research team, the horizontal deformations which have been measured geodetically by years along their horizontal bases showed little variation. It was seen, however, that the values of the horizontal deformations increased in proportion to the distance from the floor of the passage (Figure 4). Since 2012 they have amounted to −2.0%/year (sidewall base at the floor, 31–32) and −2.7%/year (sidewall base at the roof, 53–54). Taking into consideration the length of the bases, this makes for a total displacement value of about 6 mm/year or 3 mm/year for each end point of the base. It was found that the horizontal displacements of the sidewall points are half of the size of the vertical displacements of the roof benchmark.

The values of the horizontal convergence are relative and they do not allow for the determination of the participation of individual points that represent measuring baselines in the convergence process we observe. For example, we lack detailed data concerning the variation in the convergence rates and in the horizontal displacements of individual points in each month of the year. We have no data to register the impact of seasonal factors (changes in humidity being the most influential). However, the results, as has been mentioned, are essential for reasons of safety and for the provision of protection activities.

In conclusion, it can be seen that the investigations were carried out in a specific site—in the old underground workings of the Bochnia Salt Mine. Its specificity can be attributed to its tectonic position, being in the northernmost part of the Carpathians, to the geo-mechanical property of its rock salt, to its post-mining passages or chambers, and finally to the fact that it is an object within which the significance of geological and historical aspects are intertwined.

## 3. Investigations with the Application of a Sonic Probe Extensometer

The term “measurements of rock mass deformations” usually refers to displacements of control points mounted in the roof, floor or sidewalls of workings by classical surveying devices, and the obtained results actually refer to changes in their geometry. Such measurements did not, however, provide information on displacements in intact rock mass, on the spatial propagation of deformations or even provide us with any assumption for the extrapolation of these results. However, it is possible to make such observations by using a sonic probe extensometer ETM. The technical report and more details can be found in [33].

The operating principle of the extensometer is that it uses magnetic anchors and can permit the monitoring of strata movement in up to 20 locations within a section of 7.5–8.0 m. The accuracy of the equipment is to a degree of less than 0.5 mm (Figure 5). The first point of the extensometer (the magnetic anchors) is located on the extensometer’s read-out device. For our research we used 18–19 magnetic anchors. This means that we measured the displacement of 18–19 different levels in the rock salt mass.

The sonic probe extensometer implemented in the Bochnia Salt Mine has been used successfully for many years in mines [13,34,35,36,37]. In this case we are dealing with a rock mass of different geo-mechanical properties and with the amount of the expected displacements being in the order of tenths of a millimeter. In addition to this, one can expect the presence of brines migrating in the fissures and cracks which may cause disturbances in the electromagnetic field.

To determine the accuracy of the sonic probe extensometer control measurements were carried out at the Surveying Comparator Laboratory of the Cracow University of Science and Technology (UST AGH) using a Hewlett Packard precise laser interferometer, with an accuracy of ±0.02 mm. These measurements consisted of comparisons of the changes in the length of baselines in which the ends were defined both by the position of the reflectors for the interferometer measurements and by the magnetic anchors for the extensometer measurements (Figure 6). Considering the fact that its accuracy of distance measurements of laboratory baselines amounts to 0.02 mm, the results can be assumed as references which estimate very precisely the true value of the quantity of interest (baseline length). So it was assumed that the differences between observations are, practically, seen as a manifestation of the true errors of values indicated by the extensometer.

The results were analyzed as the Mean Absolute Error (MAE), which measures the average magnitude of the errors in a set of the predictions (“true” values). This is the average over the test sample of the absolute differences between the measured value (by the extensometer) and the “true” value (indications of the interferometer). So the reliability of the actual observations can be estimated according to the formula [39]:(1)$$\mathrm{MAE}=\left[n^{-1}\sum_{i=1}^{n}\left|e_{i}\right|\right]$$
where the absolute errors (*e_i_*) are derived from all individual *n* differences having equal weight.

The MAE value amounted to 0.06 mm, which is a much better result than the accuracy value reported by the manufacturer (0.5 mm).

In the next step of the analysis, absolute value was not taken (the signs of the errors were not removed), and the average error became the Mean Bias Error (MBE), which was intended to measure the average model bias. The MBE value amounted to −0.02 mm, which was even better. The obtained values are presented on Figure 7 in the form of a frequency plot. It may be seen that the results turned out to be very promising for further research.

The authors began on 19 January 2018 by measuring underground the rock mass deformations, using a sonic probe extensometer made by Trolex, at the entrance to the Chapel.

A set of three boreholes were drilled in a profile of the passage at the Chapel: in the southern side wall (S base), in the roof (ST base) and in the northern side wall (N base). They were about 7.5 m long and 45 mm in diameter. A protective tube was inserted into each of the holes; the tube is equipped with magnetic anchors that were placed along the length of the tube up until its end. These rings are stabilized in the rock mass with spring-loaded anchors. In each hole, several of such rings were placed at a distance of about 30 cm from each other. When determining this distance, the principle was followed that the greater the number and frequency of the magnetic rings in the hole, the more accurate the identification of the deformation zones in the rock mass would be. The ETM extensometer itself consists of wires placed in a 9 mm in diameter and 7 m long flexible probe. A measuring head is mounted at one of the ends, to which, during measurements, a read-out device is connected. After inserting the probe into the hole and after it is stabilized, a read-out device is connected, allowing for the measuring mode to begin. After about 30 s, the device screen displays the values of all the measuring sections in the hole. The data can be registered in the memory of the device. The measurement time for each hole should not exceed 10 min.

According to the technical data of the ETM, the reading accuracy is 0.5 mm, which gives a linear deformation value of approx. 1.6‰ for the 30 cm intervals into which each of the measurement bases (S, N, ST) is approximately divided. As has been mentioned, the laboratory test proved that the real accuracy is much better, so 0.15‰ strains should be detected. Assuming that the average annual rate of strain in the site of measurements is 2%–3‰ as that was mentioned before, it makes for the realization that an approximately one month time interval should be set as a minimal period to observe the significant changes. Finally, the measurements have been carried out at a frequency of 3 months.

## 4. Measurement Results and Discussion

The results of the probe measurements can be interpreted as displacements of individual measuring points (magnetic anchors) relative to a reference point (the “0” point). This reference point is also subjected to being displaced, meaning that the probe measurements determine the relative changes in length (depth) of the measured sections in the rock mass.

The values of the displacements were determined on the basis of the readings of the position of individual points (magnetic anchors) for the period from 2018-01-19 to 2021-03-12. In fact, the displacements were calculated in reference to the point 0, which is not stable, as the whole baselines displaces into the excavation. However, laboratory tests proved the high accuracy of the extensometer, and field surveys carried out on the condition of the salt rock mass provided some results in initial measurement series which were affected by interference. The jumps in the values of the displacements which occurred at certain points were considered to be errors of the measurement signal and were removed from further analysis.

The device readings can be applied in analysis of deformation parameters as being the basic ones: namely the displacements of each of the points of the individual baseline, and strains of the sections whose ends are defined by successive points on the particular baseline (profile). They are used here as a quantitative tool for characterizing rock salt flow field. While the displacements are calculated as being the differences in the readings of a given point position in a particular moment in time, strains in the linear deformation are calculated in accordance with the formula:(2)$$\varepsilon =\frac{{l^{\prime }}_{i}-l_{i}}{l_{i}}$$
where:*l_i_*—the length of the successive section: 0–1, 1–2, 2–3 ..., whose ends are the measuring points (locations of subsequent magnetic anchors) indicated in the 2018-01-19 measurement,*l*^′^*_i_*—the length of this section indicated in successive measurements in *i* moment of time.

The first of the two parameters have relative sense—the positions are determined in reference to the point 0 which are determined each time, as has been already mentioned, by the displacements.

However, the direction of the displacements can easily be defined as being oriented towards the shrinking excavation. The displacement rate of each individual point is controlled by the moving reference point, so an absolute value cannot be determined.

The second parameter analyzed here is strain. The strain unit is dimensionless. According to the presented formula, it is the ratio between the length shift and the initial length, so it is unitless. It reflects the effect of the mutual displacements of the points. So the calculated strains are independent of the type of system of the units, and furthermore they are independent of any reference point or coordinate system. Although they do not reflect the point of displacement itself, they give an idea of forces and stresses acting inside rock mass.

The total displacements were analyzed, which are defined here as being the difference between the initial position of each individual point (determined in the first measurement series in 19 January 2018) and its final position (determined in a particular measurement series). The changes referred to in the first results reflect a displacement process of the points (Figure 8). However, the distribution presented in the chart reflects some disturbances (jump values), but the trend is visible in all measurement baselines. After a rejection of the disturbances, linear trends were calculated and the annual rate of displacement of each point was estimated and average model-performance errors were examined.

The results are presented in Table 1. The small MAE values, with some exceptions for the last points of the roof and the southern baselines, indicate that the estimated value is very close to the measured value of a point’s displacement at a moment in time. Above all, it shows that the distribution of displacements in time is actually linear, for all points and on all baselines. However, careful analysis of the displacement rates compiled in Table 1 reveals that there are significant differences in the rate values of the individual points on each baseline. As a result, the average annual rate (AAR) estimated for the baselines are different and as follows: 3.2 mm/year for the R baseline, 1.0 mm/year for the S baseline and 1.3 mm/year for the N baseline. However, high error values were found at some points, but the displacement rates calculated for the points did not affect the average values. While the higher value of AAR for the R is not surprising (see remarks in the chapter “Environmental background of the site”), the differences between the values of AAR for the S and the N were unexpected.

The higher displacements of the points in the roof part of intact rocks surrounding the excavation correspond to observed convergence. The differentiation of the displacements observed in the southern and the northern parts of the rock mass is not justified in the mining situation, as it could be understood as being the effect of galleries or chambers that could affect the deformation process (see map in Figure 2). Moreover, the distributions of the points’ rates along the length of each baseline are quite different. These differences are outlined in Figure 9. The distributions are similar on the R and the N, but the values are different, and twice as large on the R baseline. Both of the baselines demonstrated a presence of a very even distribution with little variation in the rates along the baseline line. On the S baseline the values increase steadily and linearly with increase of the distance from the point 0. This suggests the occurrence of additional effects influencing the natural process of convergence. Summarizing, the S baseline demonstrates the displacements of the points at which average annual rate values are at their smallest, but where they are also at their most diverse. This diversity is not chaotic but presents a linear increase according to the distance from the reference point (sidewall of the excavation).

Strain analysis is an additional tool for understanding the deformation process, which from points inside the excavation reveals where there are point displacements and convergence of baselines. So the relative changes of the measured and the reference (initial) length of individual sections of each baseline were calculated. As in the case of displacements, strain values for each section (0–1, 1–2…) of each baseline in each measurement series were calculated. So for each section, temporal distribution was obtained. As with the displacement rates, the strain rates demonstrated the presence of a linear distribution.

The rate of displacement of each point was estimated and average model-performance errors were examined. The results are presented in Table 2. In contrast to the MAE values calculated for the displacement, the value of MAE calculated for the strain rates were found to be an order of magnitude smaller. The distribution of the strain rates in time for each section of each baseline is actually linear, but here we are dealing with a different distribution of the values for each individual baseline. The average annual rate (AAR) estimated for the baselines are different and they are as follows: 0.28‰/year for the R baseline, 0.08‰/year for the S baseline and 0.15‰/year for the N baseline.

As before, the highest value is represented by the R but, in opposition to the AAR of the displacements, the annual strain rates calculated for the N baseline are twice the value for the S baseline. Taking into consideration the fact that the distribution of the values on each of the baselines is not linear, it is more reliable to use the maximum values. So the maximal annual rate (MAR) estimated for the baselines are different as follows: 3.67%/year for the R baseline, 0.67%/year for the S baseline and 1.58‰/year for the N baseline. In fact it makes the same differentiation in the strain rates and the AAR and MAR values are also in a similar relationship: the values calculated for the R are twice the value for the N baseline and these are twice as much as those calculated for the S baseline. Discrepancies in the distribution of the strain rates of points on the individual baselines are outlined on Figure 10. The common feature of these distributions is the rapid change of the strain rate values, so that at the 6–7 section of each baseline (at distance of 1.5–2 m from the reference point) the values are close to zero.

Based on the results of the strain rates calculated for each point of all the baselines and assuming their spatial positions, a map of their distribution in the plane of the baselines was made. Data interpolation was performed in relation to the geometry of the radially diverging lines around the excavation. A visualization of the distribution in the form of a map, spatially oriented with respect to the excavation (the August passage), is presented on Figure 11. This visualization clearly demonstrates a strain distribution and, in fact, a stress field. It is noticeable that the southern part of the intact rocks mass surrounding the August passage is more compressed. This disturbance in the unexpected symmetry reflects presence of some additional force acting in the deformation.

The reason for this disturbance is the tectonic stress, the only stress that could be considered to be as a result of the Carpathian push in the area. The disturbance corresponds to another effect of the tectonic stress: northward inclination of the mining shafts which is discussed in the [9].

## 5. Summary and Conclusions

The presented study of deformations detected by electromagnetic extensometer measurements is presumed to be the first time research has occurred into salt mines operating in rock mass affected by tectonic stress. The obtained deformation values, which have been determined on the basis of measurements made with the use of a sonic probe extensometer ETM, were made using three 7-metre long sections placed inside the rock mass. The results showed, in general, that the values corresponded to those which had been determined with the use of geodetic measurements.

It must be noted that in this case, using the probe, the determined convergence value includes certain part of the rock mass, in which the convergence value decreases along with increasing distance from the excavation. The strain values determined in the S baseline, located in the southern sidewall, are much smaller than those found in the N baseline (the northern sidewall), which may be indicative of the presence of some additional stress direction, most probably tectonic, as there is no mining justification for this (there are no excavations that would leave an additional impact). The only stress source that could be considered is a tectonic stress resulting from the Carpathian push. Some other tectonic effects that were observed in the Bochnia Salt Mine and that were detected by surveying methods are mentioned in the text [9,40]. So the presented measurements can be used for any activities devoted to tectonics studies for conservation and future development of the Bochnia Salt Mine or any similar object.

The key significance of the work, we believe, is as follows:It is the first application of a magnetic extensometer in underground measurements for the detection of tectonic activity in the Carpathian region. The measurements proved the usefulness of the device in long-term observations in the study of tectonic activity;It is the first measurement of rock salt flow in intact rock around an underground gallery by a magnetic extensometer The results of measurements inside the rock mass can be used to calibrate numerical models in geo-mechanical analyzes of the long-term stability of workings in salt mines [41].

This type of measurement may be carried out during the mandatory periodic inspections for the maintenance of excavations, which have to take place over many years. In order to verify the obtained results and to obtain greater accuracy, it would be beneficial in future to use a continuous monitoring system using optical fibers. Such a system would also allow expansion of the range of measurements from the current distance of 7 m to a distance of up to several dozen meters.

## Figures and Tables

**Figure 1 sensors-21-05562-f001:**
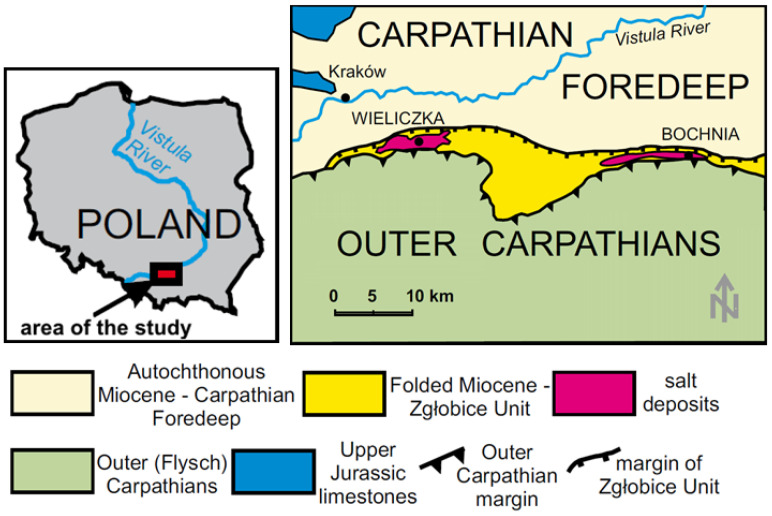
Simplified geological map and location of the Bochnia salt deposit (based on [32]).

**Figure 2 sensors-21-05562-f002:**
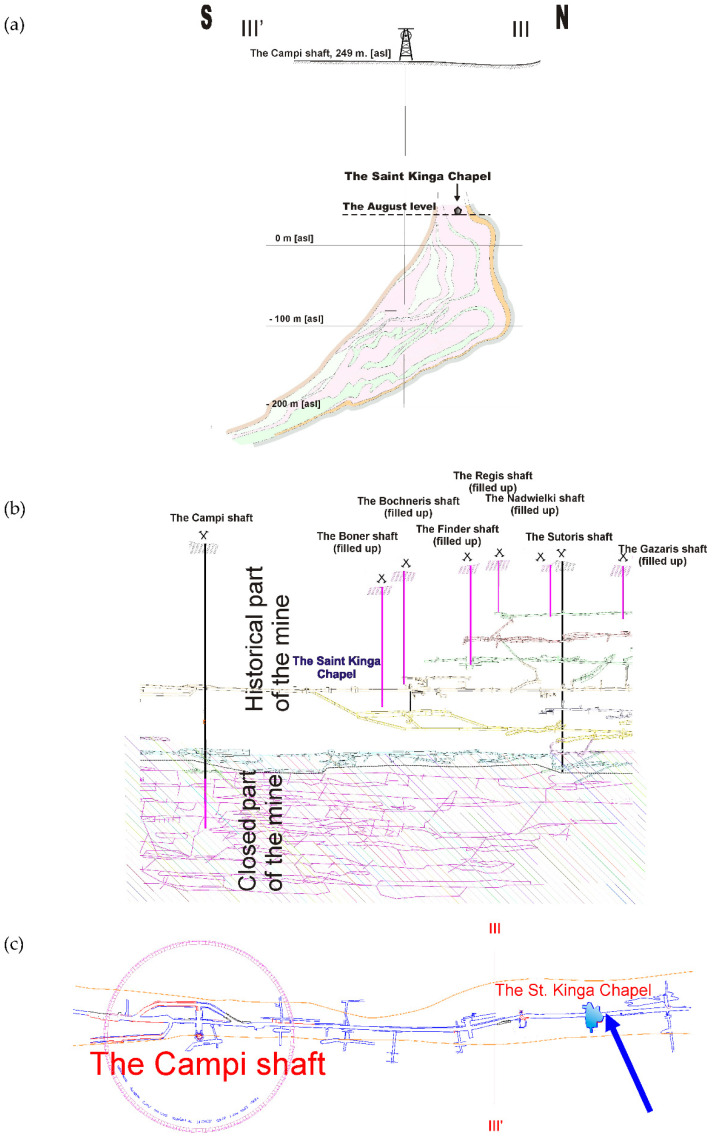
Location of the Saint Kinga Chapel (site of the measurement) in The Bochnia Salt Mine: (**a**) geological cross section of the Bochnia salt structure, (**b**) cross section illustrating the salt mine, (**c**) a situational sketch of the area of the measurements.

**Figure 3 sensors-21-05562-f003:**
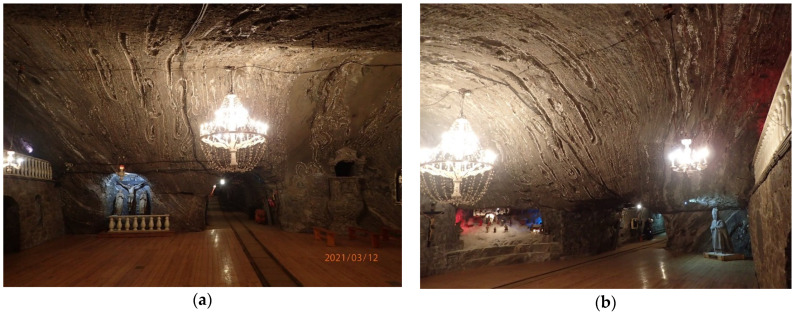
The Saint Kinga Chapel: (**a**) the eastern entrance, (**b**) the western entrance (photo by Z. Niedbalski).

**Figure 4 sensors-21-05562-f004:**
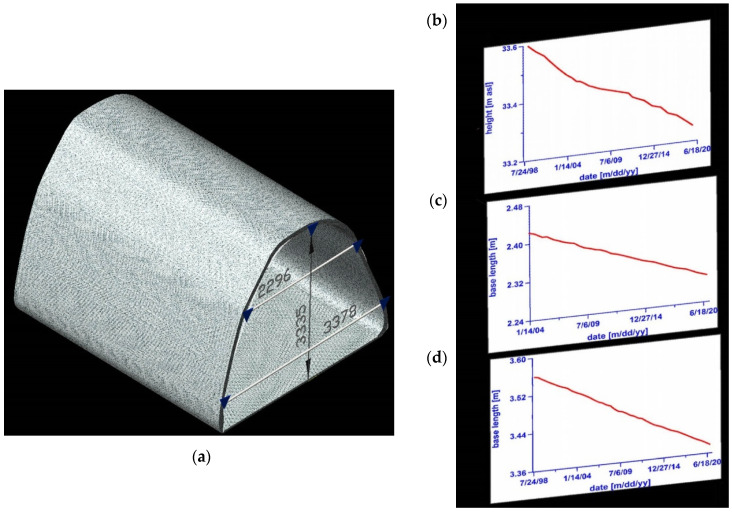
Scheme illustrating benchmarks used for displacements and convergence measurements at the entrance to the Saint Kinga Chapel. On the right: (**a**) temporal distribution of vertical displacements of the roof benchmark, (**b**) convergence of the upper base line (2296 mm), (**c**) convergence of the lower baseline (3378 mm), (**d**) convergence of height baseline (3335 mm).

**Figure 5 sensors-21-05562-f005:**
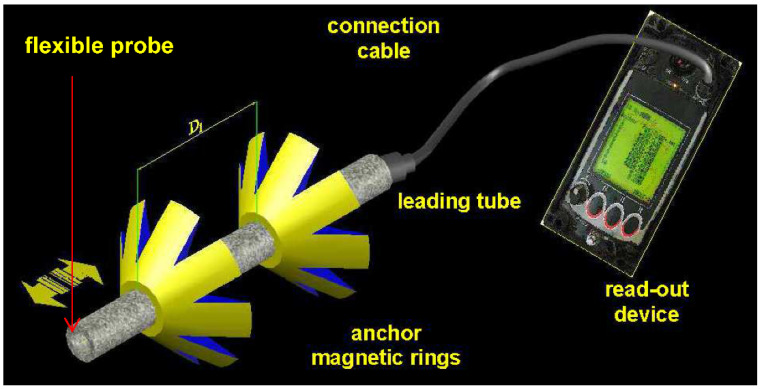
Components of the sonic probe extensometer. Graphical presentation bases on the manual [38].

**Figure 6 sensors-21-05562-f006:**
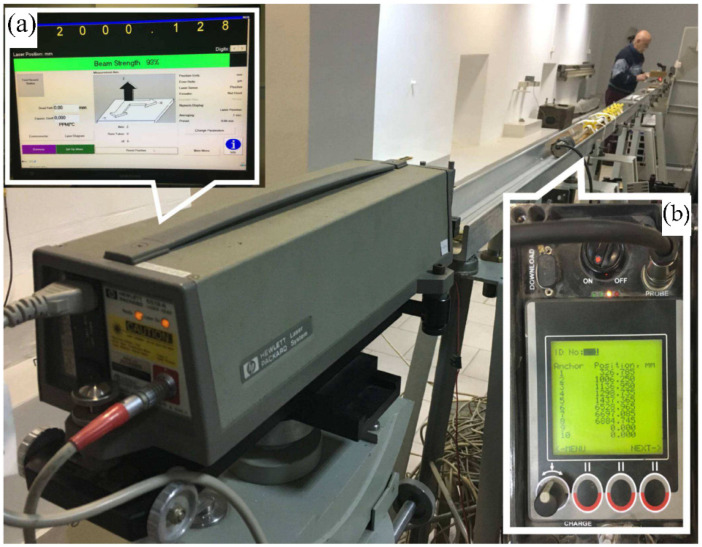
The UST AGH Surveying Comparator Laboratory. The laboratory tests measurements: using the HP interferometer: (**a**) the read-out device of the interferometer, (**b**) the read-out device of the extensometer.

**Figure 7 sensors-21-05562-f007:**
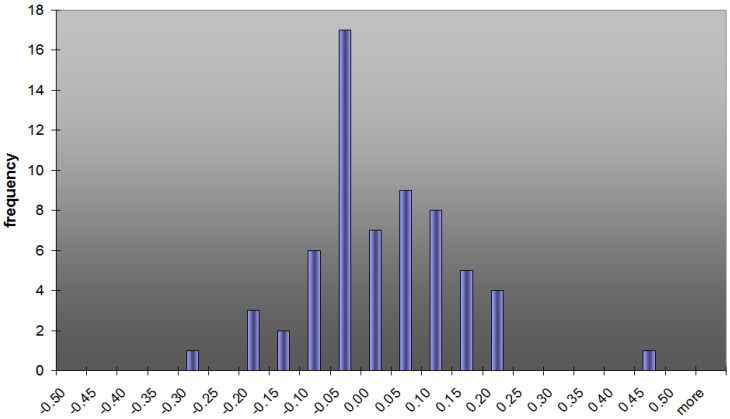
A histogram representation of the mean absolute error (MAE) of results obtained by the use of the extensometer in the laboratory tests.

**Figure 8 sensors-21-05562-f008:**
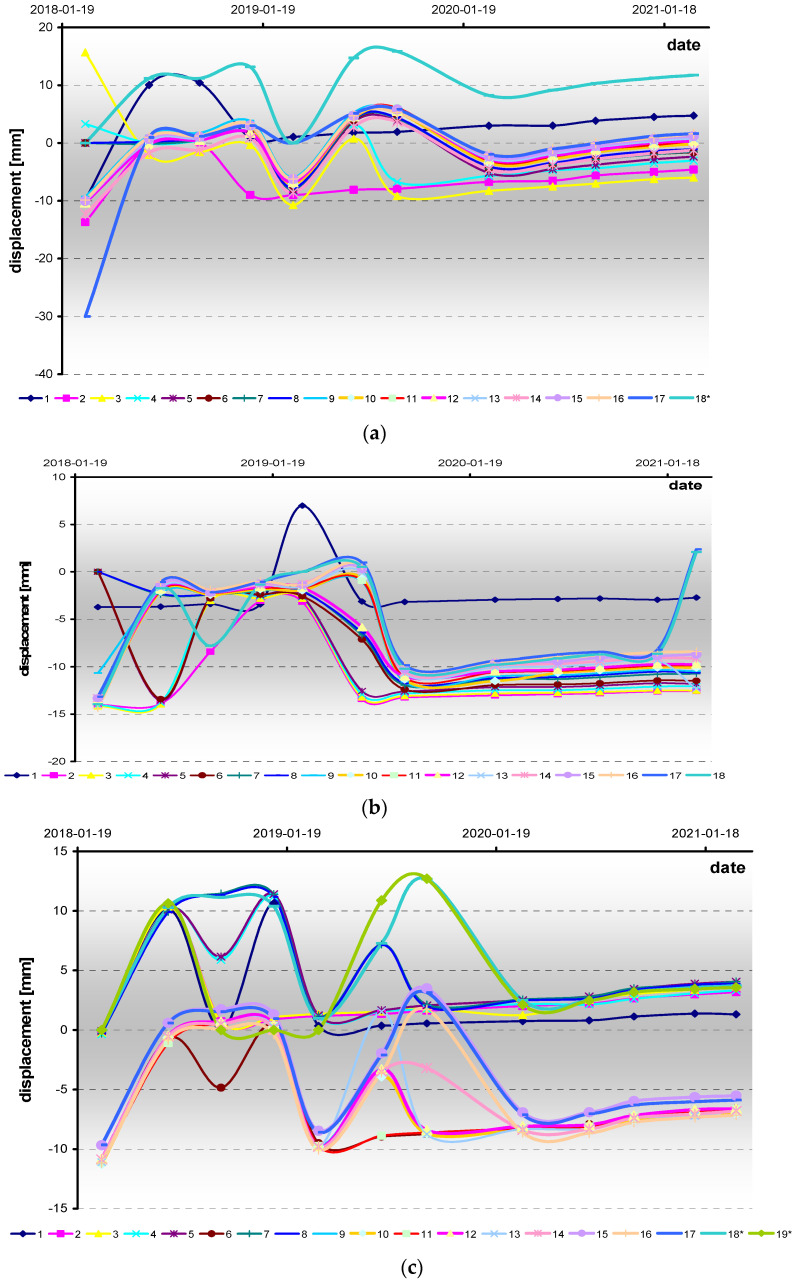
Total displacements of points since 19 January 2018 in measurement baselines: (**a**) R (roof), (**b**) S (southern side wall), (**c**) N (northern side wall). The negative values represent decreases in the distance of the measured point, and the positive represent increases in the distance. *—in reference to measurement results taken in 2 March 2018.

**Figure 9 sensors-21-05562-f009:**
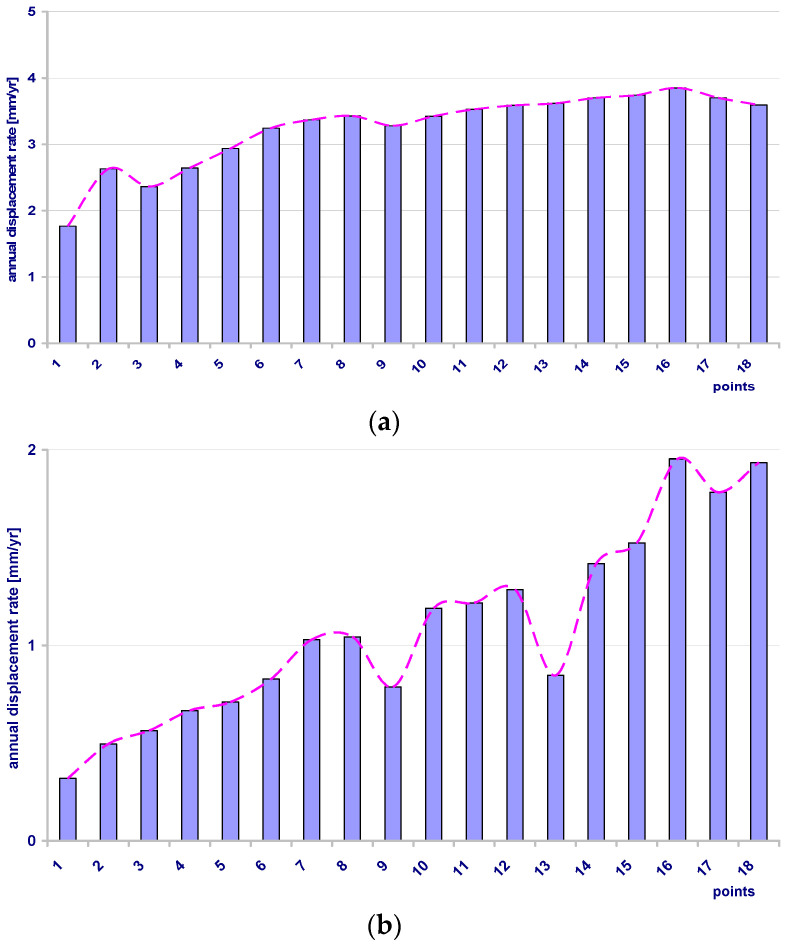
Annual displacements rates of points since 2018-01-19 in measurement baselines: (**a**) R (roof), (**b**) S (southern side wall), (**c**) N (northern side wall).

**Figure 10 sensors-21-05562-f010:**
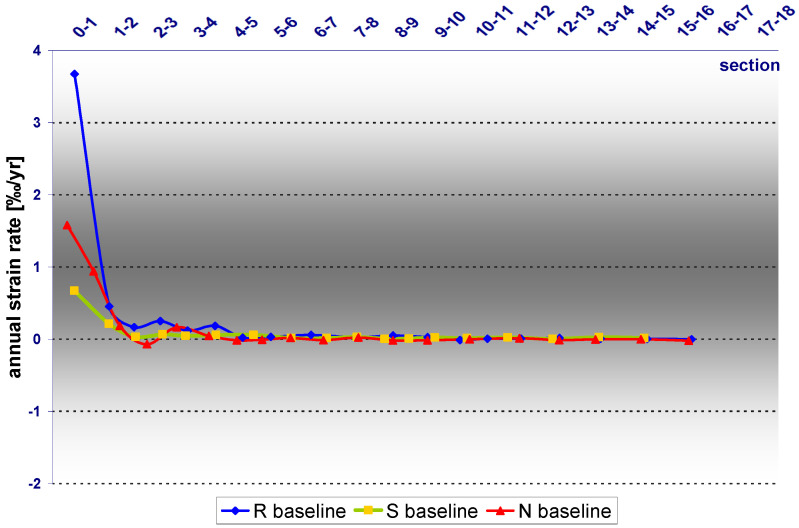
Strains calculated for particular sections of the baselines in the period 19 January 2018–12 March 2021.

**Figure 11 sensors-21-05562-f011:**
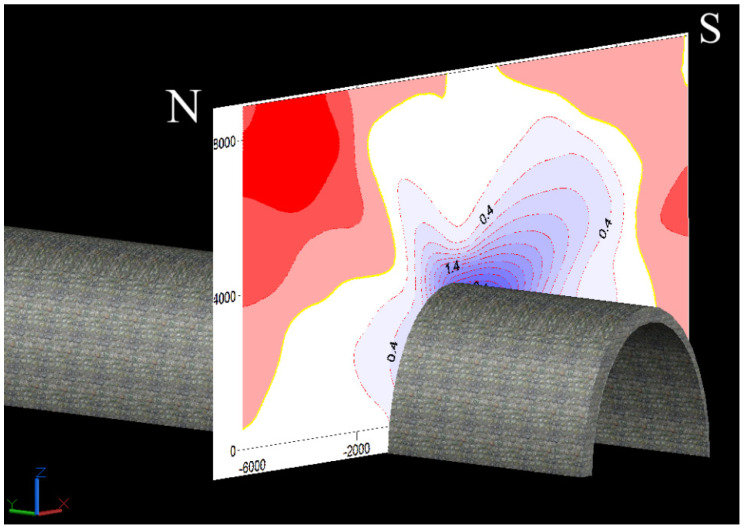
Map of strain rates determined by the extensometer measurements in aerial view.

**Table 1 sensors-21-05562-t001:** Displacements rates in measurement baselines and corresponding MAE values for R (roof), S (southern side wall), N (northern side wall).

Point	Displacement Rate for R [mm/year]	MAE for R [mm]	Displacement Rate for S [mm/year]	MAE for S [mm]	Displacement Rate for N [mm/year]	MAE for N [mm]
1	1.8	0.1	0.3	0.1	0.5	0.1
2	2.6	0.5	0.5	0.0	1.0	0.1
3	2.4	0.1	0.6	0.1	1.2	0.1
4	2.6	0.1	0.7	0.1	1.1	0.1
5	2.9	2.9	0.7	0.1	1.4	0.1
6	3.2	2.8	0.8	0.1	1.4	0.1
7	3.4	0.1	1.0	0.1	1.4	0.1
8	3.4	0.2	1.0	0.1	1.4	0.1
9	3.3	0.1	0.8	0.1	1.5	0.1
10	3.4	0.2	1.2	0.2	1.4	0.1
11	3.5	0.1	1.2	0.1	1.4	0.2
12	3.6	0.2	1.3	0.2	1.4	0.2
13	3.6	0.1	0.8	0.9	1.4	0.1
14	3.7	0.1	1.4	0.2	1.4	0.2
15	3.7	0.1	1.5	0.2	1.4	0.2
16	3.9	0.1	2.0	0.4	1.3	0.2
17	3.7	3.1	1.8	0.2	1.3	0.1
18	3.6	2.2	1.9	5.3	1.4	0.1
19	-	-			1.5	0.2

**Table 2 sensors-21-05562-t002:** Strain rates in measurement baselines and corresponding MAE values: R (roof), S (southern side wall), N (northern side wall).

Section	Strain Rate for R [mm/m/year]	MAE for R [mm/m]	Displacement Rate for S [mm/m/year]	MAE for S [mm/m]	Displacement Rate for N [mm/m/year]	MAE for N [mm/m]
0–1	3.67	0.30	1.58	0.22	1.58	0.22
1–2	0.46	0.09	0.94	0.15	0.94	0.15
2–3	0.17	0.12	0.18	0.03	0.18	0.03
3–4	0.25	0.04	−0.07	0.08	−0.07	0.08
4–5	0.12	0.09	0.16	0.10	0.16	0.10
5–6	0.19	0.05	0.05	0.03	0.05	0.03
6–7	0.02	0.04	−0.01	0.03	−0.01	0.03
7–8	0.03	0.04	−0.01	0.03	−0.01	0.03
8–9	0.06	0.05	0.02	0.02	0.02	0.02
9–10	0.03	0.05	−0.01	0.02	−0.01	0.02
10–11	0.05	0.05	0.02	0.03	0.02	0.03
11–12	0.03	0.04	−0.02	0.02	−0.02	0.02
12–13	−0.01	0.05	−0.01	0.03	−0.01	0.03
13–14	0.01	0.02	0.00	0.01	0.00	0.01
14–15	0.01	0.02	0.01	0.01	0.01	0.01
15–16	0.02	0.03	−0.01	0.01	−0.01	0.01
16–17	0.01	0.03	0.00	0.01	0.00	0.01
17–18	0.01	0.02	0.00	0.01	0.00	0.01
18–19	-	-			−0.02	0.01

## Data Availability

The datasets analyzed during the current study are available from the corresponding author on reasonable request.

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
