# Peer review of "The Application of a Sonic Probe Extensometer for the Detection of Rock Salt Flow Field in Underground Convergence Monitoring"

_sensors, 2021, doi:10.3390/s21165562_

Round 1

Reviewer 1 Report

The authors presented a very interesting topic of the application of an sonic probe extensometer for monitoring deformation of the Salt Mine in Bochnia.
In my opinion that the manuscript is well prepared and correctly written. However, the manuscript contains minor errors for revision - that should be corrected before publication.
- the summary was prepared chaotically. Each sentence covers a different topic and there is no consistency between the following information. Is the information about testing in Surveying Comparator Laboratory in the Summary necessary? In my opinion Lines 15-20 only introduce change of subject. However, the most important information is missing - the final results of measurements in the mine using the extensometer.
- keywords: please improve, now it is only a topic of the manuscript but cut into single words
- introduction: very briefly on the main topic of monitoring the condition of the mine. There is no mention or a short description of the currently used measurement methods used during such works. Please expand the text (lines 26-36) and complete the description of the introduction to the topic. There should be more bibliographies's position in this chapter in the introduction.

The following chapters have been well prepared. My only remark are the figures:
Fig. 2b, 2c - The figures show a mixture of Polish and English. Please also consider enlarging these drawings.
Fig. 5 - Reference?
Fig. 6 - too long description. And there is no information for interjected drawings, e.g. a, b

In my opinion, the conducted and presented research has been reliably conducted and correctly interpreted - I have no comments about this chapter.
- Sumary
There is no summary of the results and conclusions from the obtained work.
- Discussion ?
The manuscript lacks a discussion of the obtained results and their reference to other studies and works described in the scientific literature.
- Conclusion
There are no practical and technical recommendations for taking measurements.

In conclusion: The manuscript was very well prepared. It is a very valuable scientific and technical text. After taking into account corrections, it can be published in MDPI - Sensors.

Author Response

Dear Sir,

We read carefully your review and we would like to thank you for all your comments and remarks.

The reviewer:  The authors presented a very interesting topic of the application of an sonic probe extensometer for monitoring deformation of the Salt Mine in Bochnia. In my opinion that the manuscript is well prepared and correctly written. However, the manuscript contains minor errors for revision - that should be corrected before publication.

The authors:  thank you very much. This is very motivating.

The reviewer:  the summary was prepared chaotically. Each sentence covers a different topic and there is no consistency between the following information. Is the information about testing in Surveying Comparator Laboratory in the Summary necessary? In my opinion Lines 15-20 only introduce change of subject. However, the most important information is missing - the final results of measurements in the mine using the extensometer.

The authors:  thank you very much. We agree with this comment, so we corrected it.

The reviewer:  keywords: please improve, now it is only a topic of the manuscript but cut into single words

The authors:  We agree with this comment, so we corrected it.

The reviewer:  introduction: very briefly on the main topic of monitoring the condition of the mine. There is no mention or a short description of the currently used measurement methods used during such works. Please expand the text (lines 26-36) and complete the description of the introduction to the topic. There should be more bibliographies's position in this chapter in the introduction.

The authors:  we corrected it.

The reviewer:  Fig. 2b, 2c - The figures show a mixture of Polish and English. Please also consider enlarging these drawings.

The authors:  we corrected it.

The reviewer:  Fig. 5 - Reference?

The authors:  we added it.

The reviewer:  too long description. And there is no information for interjected drawings, e.g. a, b

The authors:  we corrected it.

The reviewer:  There is no summary of the results and conclusions from the obtained work.

The authors:  we corrected it. The text is changed.

The reviewer:  The manuscript lacks a discussion of the obtained results and their reference to other studies and works described in the scientific literature.

The authors:  we explained it in the text (lines 72-79).

The reviewer:  There are no practical and technical recommendations for taking measurements

The authors:  we added the recommendations.

The reviewer:  In conclusion: The manuscript was very well prepared. It is a very valuable scientific and technical text. After taking into account corrections, it can be published in MDPI - Sensors.

The authors:  thank you very much.

With the best regards,

Zbigniew Szczerbowski and Zbigniew Niedbalski

Reviewer 2 Report

Overall Comments

Authors have presented the results of their study on the the application of an sonic probe extensometer and the resulting measurements of the displacements in intact rocks surrounding a gallery of the Bochnia Salt Mine. Though the study is interesting, presently it is in the form of a case study or a summary of a technical report. The manuscript is unable to show and prove the novelty of this study as well.   

Some other specific comments are given below. 

Other Specific Comments

  1. Abstract is not reflective of the work done and currently it lacks the clear picture of working area
  2.  Figure 2 is not much readable.
  3. Lines 142-145: Are the vertical displacements in the roof and the floor benchmarks as mentioned in these lines are as expected? Please justify the relevancy of this statement by linking to the write up before or after this sentence.
  4. Lines 146-160: Rewrite this paragraph by correcting the English language.
  5. Lines 179-185: Rewrite this paragraph by correcting the English language.
  6. In Figure 8, what is the meaning of negative displacements? This needs to be described in the text.
  7. Result and discussion section is poorly described and not clearly understandable. It also lacks the comparison of the similar studies in the past and the novelty of this study.

Author Response

Dear Sir,

We read carefully your review and we would like to thank you for all your comments and remarks.

The reviewer:  Authors have presented the results of their study on the the application of an sonic probe extensometer and the resulting measurements of the displacements in intact rocks surrounding a gallery of the Bochnia Salt Mine. Though the study is interesting, presently it is in the form of a case study or a summary of a technical report. The manuscript is unable to show and prove the novelty of this study as well.

The authors:  thank you very much. We agree with this comment, so the text is expanded, especially the Summary paragraph. We show now the novelty of this study.

The reviewer:  Abstract is not reflective of the work done and currently it lacks the clear picture of working area

The authors:  thank you very much. We agree with this comment, so we corrected it.

The reviewer:  Figure 2 is not much readable.

The authors:  We corrected it.

The reviewer:  Lines 142-145: Are the vertical displacements in the roof and the floor benchmarks as mentioned in these lines are as expected? Please justify the relevancy of this statement by linking to the write up before or after this sentence.

The authors:  Yes. They correspond to other result of measurements in this part of the mine (all benchmarks in this part of the mine have been showing the same displacements by years).

We added explanations in the text (lines 193-195).

The reviewer:  Lines 146-160: Rewrite this paragraph by correcting the English language.

The authors:  we corrected it.

The reviewer:   179-185: Rewrite this paragraph by correcting the English language.

The authors:  we corrected it.

The reviewer:  In Figure 8, what is the meaning of negative displacements? This needs to be described in the text.

The authors:  The negative values represent decrease of the distance of the measured point, the positive – the increase of the distance. We put this explanation into the text.

The reviewer:  Result and discussion section is poorly described and not clearly understandable. It also lacks the comparison of the similar studies in the past and the novelty of this study.

The authors:  we corrected it.

With the best regards,

Zbigniew Szczerbowski and Zbigniew Niedbalski

Reviewer 3 Report

The application of an sonic probe extensometer for the detection of rock salt flow field in underground convergence monitoring
Zbigniew Szczerbowski, Zbigniew Niedbalski

  1. line 39-41 miss a general reference about mining plants.
  2. line 50-56 miss a general reference.
  3. line 57-68 miss a general reference.
  4. line 69-76 miss a general reference.
  5. Also, citations from recent papers are required.
       6 . Figure 2 ((b) plan of the salt mine, (c) the place of measurements), the figure quality are poor, which makes it difficult to read. Please, redraw the figure (b) and (c).
  1. There are no references in the 4. Measurement results and discussion part.
  2. I found this article interesting and indeed highlighting a knowledge gap of importance. However, there are many errors about formatting and typo. It would be good to first correct the formal error and then look at the paper.
  3. Conclusions part is too short. Please, rewrite the conclusion part more specifically.

Author Response

Dear Sir,

We read carefully your review and we would like to thank you for all your comments and remarks.

The reviewer:   line 39-41 miss a general reference about mining plants.

The authors:  We corrected it.

The reviewer:  line 50-56 miss a general reference.

The authors:  We corrected it.

The reviewer: line 57-68 miss a general reference..

The authors:  We corrected it.

The reviewer:  line 69-76 miss a general reference.

The authors:  We added it.

The reviewer:  Also, citations from recent papers are required..

The authors:  We added citations and the number of references expanded from 27 to 42 positions.

The reviewer:   Figure 2 ((b) plan of the salt mine, (c) the place of measurements), the figure quality are poor, which makes it difficult to read. Please, redraw the figure (b) and (c)..

The authors:  we corrected it.

The reviewer:  There are no references in the 4. Measurement results and discussion part.

The authors:   Most of the chapter is just a description of measurements and elaboration of the result. Anyway we added a reference related to the obtained results .

The reviewer:   I found this article interesting and indeed highlighting a knowledge gap of importance. However, there are many errors about formatting and typo. It would be good to first correct the formal error and then look at the paper.

The authors:  we corrected it.

The reviewer:  Conclusions part is too short. Please, rewrite the conclusion part more specifically.

The authors:  we agree with this comment, so the paragraph was expanded.

With the best regards,

Zbigniew Szczerbowski and Zbigniew Niedbalski

Round 2

Reviewer 2 Report

Authors have addressed most of my concerns. Thank you very much. Looking at the moderate scientific value of this paper, I recommend this article to accept.

Reviewer 3 Report

After the first revision round, the revised article has improved. The paper is interesting and worthy of publication.